# Effect of Drying on Lettuce Leaves Using Indirect Solar Dryer Assisted with Photovoltaic Cells and Thermal Energy Storage

Pedro Cerezal Mezquita [1,2,*], Aldo Álvarez López [1] and Waldo Bugueño Muñoz [1,2]

[1] Departamento de Ciencias de los Alimentos y Nutrición, Facultad de Ciencias de la Salud, Universidad of Antofagasta. Avenida Universidad de Antofagasta # 02800, Campus Coloso, Casilla 170, Antofagasta 1240000, Chile; aldoalvarez1984@icloud.com (A.Á.L.); waldo.bugueno@uantof.cl (W.B.M.)

[2] Centro de Desarrollo Energético Antofagasta (CDEA), Facultad de Ingeniería, Universidad de Antofagasta, Avenida Universidad de Antofagasta # 02800, Campus Coloso, Casilla 170, Antofagasta 1240000, Chile

* Correspondence: pedro.cerezal@uantof.cl; Tel.: +56-55-2637717

**Abstract:** The thin layer drying behavior of lettuce leaves was investigated using an indirect pilot solar dryer with thermal energy storage in water, equipped with solar collectors and photovoltaic cells. The drying procedure consisted of shredded lettuce leaves, temperature ≤ 52 °C, airspeed, 1.0 m·s$^{-1}$, and process time ~10.0 h. Fifteen drying models were adjusted to the experimental data obtained; three models with maximum values of coefficient of determination ($R^2$)—Page, Midilli, and Kucuk, and Weibull Distribution, whose values of $R^2 \geq 0.998$, and other statistical parameters, $\chi^2$, SSE, and RMSE values closer to zero were chosen. The initial browning index BI = 120.5 ± 0.7 decreased compared to the dry sample BI = 78.99 ± 0.5, with chromatic coordinate degradations a* and b*; but not the luminosity L*; where ΔE = 8.26; whose meaning is that the dry sample is a "more opaque brownish color" due to the difference in the chroma ΔC = 6.65, and with a change from the yellow-green to yellow-red zone, and a difference in hue angle, Δh° = 14.27, between the fresh and the dried sample. D$_{eff}$ values for shredded lettuce leaves were $1.8 \times 10^{-9}$ m$^2$ s$^{-1}$ for values ≤ 52 °C.

**Keywords:** mathematical modeling; thin layer drying; solar drying; lettuce leaves

## 1. Introduction

The post-harvest loss in vegetables has been estimated to be about 30–40% due to inadequate post-harvest handling, lack of infrastructure, processing, marketing, and storage facilities [1]. Therefore, the food processing sector can play a vital role in reducing the post-harvest losses by processing and value addition of vegetables, which will ensure better remuneration to the growers [2].

Drying of foods is a unit operation used for moisture removal by application of heat in controlled conditions. It involves the simultaneous transfer of heat, mass, and momentum, where the heat penetrates into the food to evaporate the water, which is removed by circulating unsaturated gaseous phase or air drying [3]. In simple terms, the technique of dehydration requires the development of methods to minimize the adverse effects of treatment [4,5]. Advances in technology allow for a better quality of dried products and can be controlled with parameters such as velocity and temperature of drying air, consequently decreasing the operating time [6].

Drying is a complex unit operation to model mathematically, and there have been a number of continuum type mechanisms proposed and the associated mathematical models established. These include liquid diffusion, capillary flow, dual (temperature, water content gradient) and triple (temperature, water content and pressure gradient) driving force mechanisms,

evaporation-condensation, another dual driving force mechanism, and dual-phase (liquid and vapor) transfer mechanism [7].

In drying processes, it is assumed that diffusivity, interpreted with the Fick diffusion equation, is the only physical mechanism for transporting water to the surface of the product. Moreover, to the limited information on the mechanism of moisture movement during drying and the complexity of the process, the effective moisture diffusivity is affected by moisture content, temperature, and porosity of the material [8].

Lettuce (*Lactuca sativa* L.) is a major leafy vegetable and is commonly used in salad mixtures and sandwiches, has great importance in the human diet, and is perhaps the world's most important in fresh salad greens [9]. Its nutritional characteristics include that it is a vegetable with high moisture, essential nutrients, and of low-calorie content [10].

Lettuce (*Lactuca sativa* L. var. longifolia), corresponds to lettuce plants known as "Romaine". This vegetable plant has large leaves that are erect and oblong, each being 20–30 cm long and 6–10 cm wide. It has a prominent midrib, slightly undulating surface, and irregularly toothed edges. The stem is longer than the other cultivars and remains protected by the leaves, forming a cylindrical or conical head [11].

Romaine or cos lettuce is a variety of lettuce (*Lactuca sativa* L. var. longifolia) that grows in a tall head of sturdy leaves with firm ribs down their centers. Unlike most lettuces, it is tolerant of heat [12]. In Great Britain, Romaine lettuce is known as "cos lettuce" [13]. Many dictionaries trace the word cos to the name of the Greek island of Cos, from which the lettuce was presumably introduced. Other sources trace it to the Arabic word for lettuce, "*khus*"[14].

Fresh-cut fruits and vegetables generally have a short shelf life and may show tissue browning as a consequence of stressful conditions that occur during post-harvest processing (handling, cutting, packaging, etc.) and storage [15]. Tissue browning is a typical disorder of fresh-cut lettuce [16]. The results obtained in studies conducted with fresh-cut leaves of two lettuce cultivars suggest the occurrence of tissue browning is presumably because of a conversion of ascorbate to non-active forms [17], which contrast the conclusions reported by Cantos et al. [18], which indicated the effects of wounding on phenolic enzymes in six minimally processed lettuce cultivars upon storage.

On the other hand, the use of agricultural by-products and wastes derived from the food industry and those generated by the wholesale markets are largely made up of plant leaves. Its accumulation is a growing environmental problem. However, their price and availability make them reusable for animal feed and, therefore, are of great importance [19].

There are no reports in the scientific literature describing drying lettuce leaves in solar dryers, except one paper presented by Rojano-Aguilar et al. [20] that collects a preliminary experience with a solar device with basic characteristics and testing the possible fit of 16 thin-layer drying models, determining that the best fitting of the experimental drying data was obtained with the Two Terms model.

The aim of this study was focused on the drying of shredded lettuce (*Lactuca sativa* L. var. longifolia) leaves, known in Chile as "Costinas" in a pilot solar dryer of indirect operation using thermal energy storage in water and equipped with solar collectors (U-pipe) for converting the solar irradiance in heat energy, and photovoltaic cells (monocrystalline) for converting the solar irradiance in electricity. The drying kinetics of the lettuce were studied using different thin-layer drying models.

## 2. Materials and Methods

### 2.1. Material

Romaine lettuce (*Lactuca sativa* L. var. longifolia) leaves were selected from healthy and uniform leaves used for the drying experiments. They were stored at 8.0 ± 2.0 °C, prior to the drying process.

The initial moisture determination of the shredded lettuce leaves sample was performed by drying on infrared balance, model ML-50 (A&D, Moisture Analyzer, Tokyo, Japan). The initial amount used

in each analysis was 5 g, and it was concluded until a constant weight was reached. The analyses were carried out in triplicate at 100 °C.

## 2.2. Solar Drying Equipment

The experimental equipment mainly consists of a pilot solar dryer of indirect operation using thermal energy storage in water and equipped with solar collectors (U-pipe) for converting the solar irradiance in heat energy and photovoltaic cells (monocrystalline) for converting the solar irradiance in electricity (Figure 1). This equipment has two drying chambers that operate in series, and each has three trays to put samples on. Each chamber has a digital balance to record the weight loss of the samples placed in the three trays inside.

Heating mode of the air in the solar dryer: The hot water stored in the hydro-pneumatic tank with isothermal conditions is pumped to an air-water exchanger whose function is to heat the air. Then, the hot air passes into drying chamber 1, and at the end of its route, it goes to drying chamber 2. At the end of its route through the chambers, the air with part of the product's water continues its passage through a rectangular aluminum duct where it exchanges heat with fresh air coming in through the intake from the outside. This last stage of circulation of hot exhaust air aims to deliver some of its heat to the air at room temperature that is entering the air-water exchanger, allowing energy savings.

## 2.3. Experimental Procedures

The drying evaluations took into account the following considerations: (a) lettuce var. longifolia as vegetable leaves with high humidity, (b) exposure to dry form: shredded; (c) constant velocity of the drying air, 1.0 m·s$^{-1}$; (d) inlet drying air temperature $\leq$ 60 °C; the drying process was conducted using the pilot solar dryer in the beginning summer in the southern hemisphere, December 2016–January 2017, with air temperatures in the environment between 17.0 and 23.0 °C over a one-day drying cycle, under relative humidity between 60.5 and 82.7%, with solar radiation changing between 8.25 and 8.74 kWh/m$^2$ in Antofagasta, Chile. In each of the drying chambers of the equipment, 350 to 440 g in weight of fresh shredded lettuce leaves were placed and uniformly distributed in the three trays at the beginning of the drying process, and the thickness of the thin layer was less than 10 mm. Every 20 min during the drying period, the weight of the three trays in each drying chamber was determined using a digital balance (0.05 g precision) (Radwag, WLC model 20/A2, Radom, Poland) coupled to them. The drying process was continued until the moisture content remained constant.

## 2.4. Drying Curves

For the determination of the drying curves, it was necessary to record the data obtained experimentally at constant time intervals (minutes) with which it is possible to calculate the moisture of the product (g water/g dry solid) at each time and to obtain the graphs relating moisture loss over time. Thus, in the experimental runs conducted on the shredded lettuce, the amount of water evaporated during the drying process was determined directly from the data obtained in the drying chambers in the solar drying facility every 20 min. Drying tests were replicated on three occasions between the months of December 2016 and January 2017. Each experiment started between 9:30–10:00 h and concluded when the product lost moisture and reached constant weight approximately between 19:30 and 20:00 h.

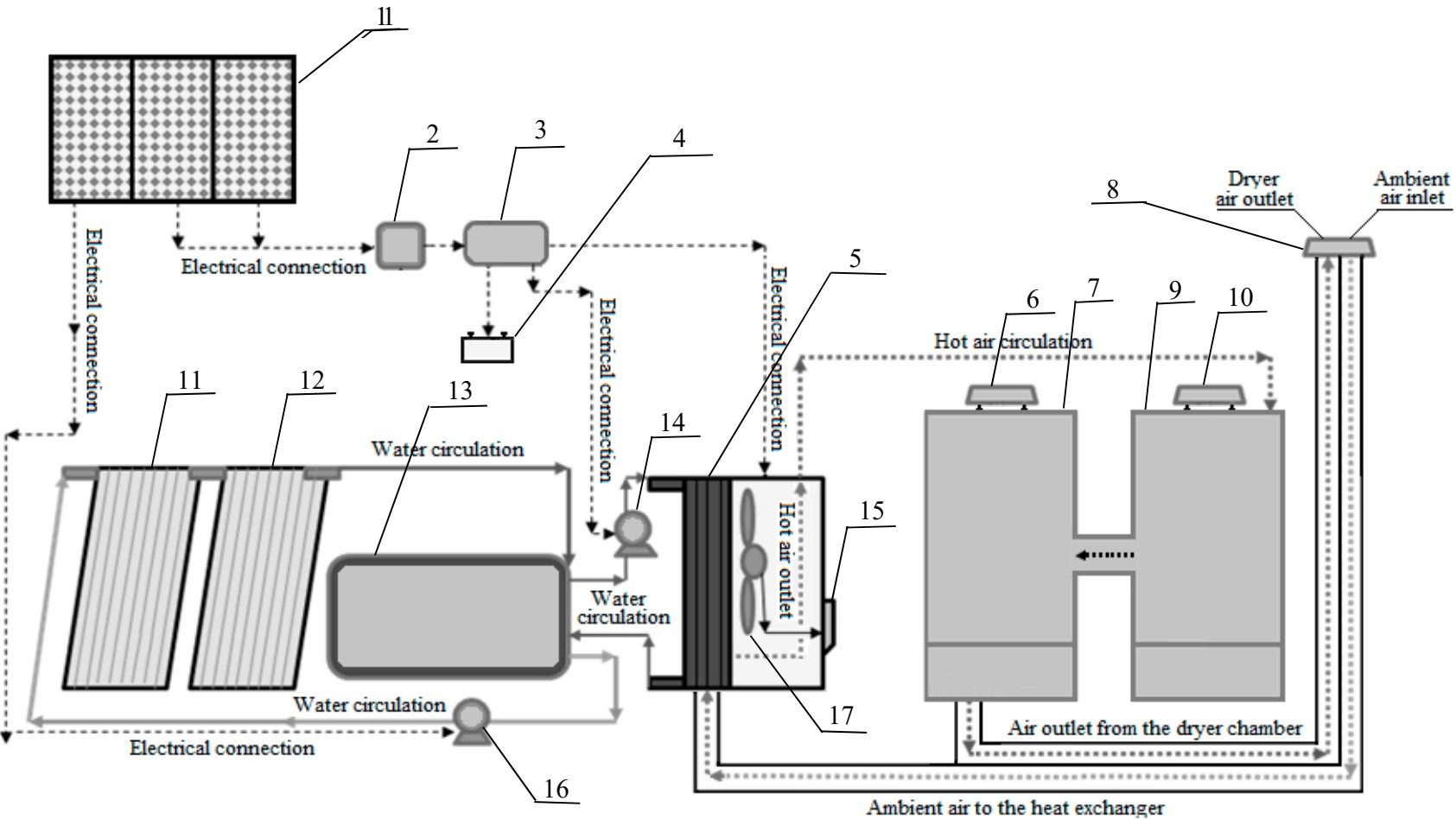

**Figure 1.** Schematic representation of the indirect solar dryer assisted with photovoltaic cells and with the thermal energy storage unit. (1) Three photovoltaic cells, (2) voltage controller, (3) electrical switches board, (4) battery, (5) heat exchanger air-water, (6) digital balance, (7) drying chamber # 2, (8) window for entry and exit of air dryer, (9) drying chamber # 1; (10) digital balance, (11) (12) solar collectors for water (U-pipe), (13) hydro-pneumatic tank water with inner membrane for thermal energy storage (with external insulation), (14) water pump, (15) variable speed drive for airflow, (16) water pump, (17) variable speed fan.

*2.5. Calculations for Determining the Drying Curves and Drying Rate*

Drying kinetics were constructed by plotting the moisture ratio on a dry basis (db) at any time (*M*) divided by the initial moisture (db) ($M_o$), both given in grams, which is known as moisture ratio ($\frac{M}{M_o}$) versus time. The moisture content (g water/g dry solids) was determined using the following equation:

$$M = \frac{(W_o - W) - W_1}{W_1}$$
(1)

where *M*: the moisture content at any time (g water/g dry solid); $W_o$: the initial weight of sample (g); *W*: amount of evaporated water (g); $W_1$: the dry matter content of the sample (g). The moisture ratio (*MR*) was simplified to ($\frac{M}{M_o}$), being $M_o$ initial moisture content in the initial time = 0 in place of $\frac{(M-M_e)}{(M_o-M_e)}$, for mathematical modeling of the solar drying curves due to the continuous fluctuation of the relative humidity of the drying air during the solar drying process [21].

*2.6. Modeling of Drying Curves*

Fifteen drying models were fitted to the experimental data obtained: Three models derived from Newton's law of cooling; nine models derived from Fick's second law of diffusion, and three empirical models [22]. The detailed review of thin-layer drying equations are shown in Table 1.

**Table 1.** Mathematical models applied to the drying curves.

| N° | Model Name | Model | References |
|----|------------|-------|------------|
| *Models derived from Newton's law of cooling* | | | |
| 1 | Newton | $MR = exp\,(-k\,t)$ | [23] |
| 2 | Page | $MR = exp\,(-k\,t^{\,n})$ | [24] |
| 3 | Modified Page I | $MR = exp\,(-k\,t)^n$ | [25] |
| *Models derived from Fick's second law of diffusion* | | | |
| 4 | Henderson and Pabis | $MR = a\,exp\,(-k\,t)$ | [26] |
| 5 | Modified Henderson and Pabis. | $MR = a\,exp\,(-k\,t) + b\,exp\,(-g\,t) + c\,exp\,(-h\,t)$ | [27] |
| 6 | Logarithmic | $MR = a\,exp\,(-k\,t) + c$ | [28] |
| 7 | Approximation of diffusion | $MR = a\,exp\,(-k\,t) + (1-a)\,exp\,(-k\,b\,t)$ | [28] |
| 8 | Midilli and Kucuk | $MR = a\,exp\,(-k\,t^{\,n}) + b\,t$ | [29] |
| 9 | Two Term | $MR = a\,exp\,(-k_0\,t) + b\,exp\,(-k_1\,t)$ | [30] |
| 10 | Two Term exponential | $MR = a\,exp\,(-k\,t) + (1-a)\,exp\,(-k\,a\,t)$ | [31] |
| 11 | Aghbashlo Model | $MR = exp\,(-(k_1\,t)/(1 + k_2\,t))$ | [32] |
| 12 | Verma Model | $MR = a\,exp(-k\,t) + (1-a)\,exp(-g\,t)$ | [33] |
| *Empirical models* | | | |
| 13 | Wang and Sing | $MR = 1 + a\,t + b\,t^2$ | [34] |
| 14 | Thompson | $MR = exp\,((-a - (a^2 + 4\,b\,t)^{0.5})/2b)$ | [35] |
| 15 | Weibull Distribution | $MR = a - b\,exp\,(-k\,t^{\,n})$ | [36] |

MR: Moisture ratio; k: kinetic parameter (min$^{-1}$), n and a: empirical parameters (dimensionless); t: drying time (min); b, c, g, h, $k_o$, $k_1$, and $k_2$: empirical constants in the drying models [37].

*2.7. Color Analysis*

Color measurements of the samples of shredded fresh lettuce, prior to the drying process and after the drying process were performed by using the CIE *L\*a\*b\** (CIELAB), since it is the most complete color space specified by the International Commission on Illumination (Commission International d'Eclairage, hence its CIE initialism); following the operating instructions provided by the manufacturer, Hunter Colour Instruments (ColorFlex Hunterlab, VA, USA), the recommended illuminant/observer: D65/10°, using the method of measurement for opaque or translucent solids samples in powder. The parameters *L\** (measures the sample's lightness; 0 = black; 100 = perfect white), *a\** measures redness (+*a\** = red, −*a\** = green), meanwhile *b\** related to yellowness (+*b\** = yellow, −*b\** = blue) were

registered. Ten measurements of each sample were made, and each was done in triplicate every 1 min. The equations of color measures were as follows:

Color difference $\Delta E$ was calculated by the equation:

$$\Delta E = \sqrt{(L_F^* - L_D^*)^2 + (a_F^* - a_D^*)^2 + (b_F^* - b_D^*)^2} \tag{2}$$

where subscript F stands for the color reading of fresh lettuce, and the parameters with subscript D, refer to color values at the end of drying.

With the data obtained from the coordinates $L^*$, $a^*$, and $b^*$, the ratio of redness over yellowness ($R$), hue angle ($h^o$) and Chroma ($C$) were calculated according to:

$$R = \frac{a^*}{b^*} \tag{3}$$

$$h^o = tan^{-1}\left(\frac{b^*}{a^*}\right) \tag{4}$$

$$C = \left(a^{*2} + b^{*2}\right)^{1/2} \tag{5}$$

As well as Browning index ($BI$) as:

$$BI = \frac{[100\,(x - 0.31)]}{0.17} \tag{6}$$

where:

$$x = \frac{(a^* + 1.75\,L^*)}{(5.645\,L^* + a^* - 3.012\,b^*)} \tag{7}$$

Browning index ($BI$) represents the purity of the brown color and is considered as an important parameter associated with browning in processes where enzymatic and non-enzymatic browning takes place [38].

### 2.8. Calculation of Effective Moisture Diffusivity

Effective moisture diffusivity of shredded lettuce leaves was estimated from the analytical solution of Fick's second equation, considering a constant moisture diffusivity, infinite slab geometry, and uniform initial moisture distribution [39] is stated as:

$$MR = \frac{8}{\pi^2} \sum_{n=0}^{\infty} \frac{1}{(2n+1)^2} exp\left(-\frac{(2n+1)^2\,\pi^2 D_{eff}\,t}{4L^2}\right) \tag{8}$$

where $D_{eff}$ is the effective diffusivity (m$^2$·s$^{-1}$), L is the half-thickness of the lettuce leaves ($L = 0.00075$ m), and $n$ is a positive integer, also called the Fourier's series number. Simplified by taking the first term of the series solution:

$$MR = \frac{8}{\pi^2} exp\left(-\frac{\pi^2 D_{eff}\,t}{4L^2}\right) \tag{9}$$

The diffusion coefficient for drying can be calculated from the slope obtained by plotting the natural logarithm of $MR$ against the drying time.

$$ln\,(MR) = ln\left(\frac{8}{\pi^2}\right) - \left(\frac{\pi^2 D_{eff}\,t}{4L^2}\right) \tag{10}$$

Effective diffusivity is also typically calculated by using the slope of Equation (10), namely when the natural logarithm of *MR* versus time is plotted, a straight line with a slope *K* is obtained:

$$K = \frac{\pi^2 \, D_{eff}}{4L^2} \tag{11}$$

$D_{eff}$ value was calculated from Equation (11) as follows Equation (12):

$$D_{eff} = \frac{4L^2 \, K}{\pi^2} \tag{12}$$

*2.9. Statistical Analysis*

Statistical parameters, $R^2$, SSE, RMSE and, $\chi^2$, were used to determine the goodness of fit of the drying curves of the fifteen models studied. The determination coefficient ($R^2$) was the primary criterion for selecting the most suitable equation to describe the drying curves of shredded lettuce leaves in the solar dryer [40]. In addition, other statistical parameters were used to compare the goodness of fit of the drying models with the experimental data, these being the standard error of estimated (SEE), which provides information on the long-term performance of the correlations by allowing a comparison of the actual deviation between predicted and measured values term by term; the root mean square error (RMSE) which provides information on the short term performance and, reduced the Chi-square ($\chi^2$) is the mean square of the deviations between the experimental and predicted moisture levels. The closest values to 1.0 for $R^2$ and those closest to zero for SSE, RMSE, and $\chi^2$ are commonly regarded as the optimal criterion to evaluate the goodness of fit of the models used [28,41,42]. The Equations (13)–(16) for these statistical parameters are:

$$R^2 = \frac{\sum_{i=1}^{N} \left( MR_{exp,i} - MR_{exp_{mean,i}} \right)^2 - \left( MR_{pre,i} - MR_{exp,i} \right)^2}{\sum_{i=1}^{N} \left( MR_{exp,i} - MR_{exp_{mean,i}} \right)^2} \tag{13}$$

$$SSE = \frac{1}{N} \sum_{i=1}^{N} \left( MR_{exp,i} - MR_{pre,i} \right)^2 \tag{14}$$

$$RMSE = \left[ \frac{1}{N} \sum_{i=1}^{N} \left( MR_{exp,i} - MR_{pre,i} \right) \right]^{1/2} \tag{15}$$

$$\chi^2 = \frac{\sum_{i=1}^{N} \left( MR_{exp,i} - MR_{pre,i} \right)^2}{N - n} \tag{16}$$

where $MR_{exp,i}$ stands for the experimental moisture ratio found in any measurement; $MR_{pre,i}$ predicted moisture ratio for this measurement; *N*, the total number of observations; *n*, number of constants [40,42].

In addition to the above criteria, other requirements of the statistical parameters for the selection of the best drying models were established, these being $R^2 \geq 0.998$; $\chi^2 \leq 0.00025$, SSE $\leq 0.00020$, and RSME $\leq 0.0140$.

For adjusting the best mathematical model describing the thin layer drying curve of shredded lettuce samples, those fifteen models of non-linear regression presented in Table 1 were used. Non-linear regression analysis was performed by using the software Statgraphics Centurion (v.15) to estimate the parameters of equations of Table 1.

All sample analyses were performed at least in triplicate; appearing as mean values ($\overline{X}$) and their respective standard deviations (*S*). Each of the statistical analyses was performed with a confidence level of 95%. One or two-way analysis of variance (ANOVA) with subsequent comparison by the Duncan multiple range tests were performed according to the requirements [43].

## 3. Results and Discussion

### 3.1. Drying Characteristics

3.1.1. Drying Curve

The mean values of the three drying runs made with shredded lettuce leaves are shown in Figure 2. It can be seen that the moisture ratio reduced exponentially as the drying time increased but indicates no constant rate in the drying. This was similar to the reported results from other herbs and food materials, showing a rapid moisture removal in the initial stage that later decreased with the drying time [44,45]. This continuous decrease in the moisture ratio, as the processing time elapsed, is an indicative factor that the diffusion mechanism has governed the internal mass transfer [46]. The initial moisture content of the lettuce leaves was 94.48% ± 0.11, being the required drying time to reach the mean final moisture contents (d.b.) ≤ 10% for three drying runs ≈ 10 h. These results are in accordance with the results obtained when dried using open sun and shade for Vernonia amygdalina leaves [47], in thin-layer drying of parsley and lettuce leaves, other vegetables leaves, and some aromatic plants also in the open sun [20,46,48,49], as well as in the thin layer solar drying of an indirect-mode forced-convection solar dryer, used for drying Thymus and mint leaves [50].

The drying rates were low during the first 60 min of the process because the air temperature had mean values of 35.2 and 32.2 °C in those instances for the first and second drying chambers, respectively (Figure 2). This is the logical operation of this type of dryer for the first stage, since the water from the storage tank, not yet reached a high temperature to allow the incoming air as it passes through the heat exchanger and get hot enough. At this stage, the collector by which the water passes, receives a relatively low solar incidence.

Higher values of the air temperature in the first and second drying chambers were 51.7 and 47.6 °C at the time 400 min, for the three drying runs performed, approximately between 4:00 p.m. and 4:30 p.m. (Figure 2). This was due to the fact that the highest temperatures of the inlet and recirculation of water to the heat exchanger occurred between 280 and 400 min with mean values of 57.5 to 58.9 °C and 56.6 to 58.1 °C, respectively (Figures S1 and S2, Supplementary Materials). In the last stage of the drying process, decreases in drying air temperatures were observed from 44.4 to 43.2, and 43.5 to 41.9 °C in 560 and 600 min, for chambers 1 and 2, respectively (Figure 2); caused by the decrease in inlet and recirculation water temperatures in the heat exchanger from 48.3 to 47.3, and 47.2 to 46.9 for 560 and 600 min, respectively (Figures S1 and S2, Supplementary Materials).

3.1.2. Drying Rate Curve

Figure 3 shows the mean values of the drying rate as a function of the moisture content during the whole process according to the temperatures of the drying air to shredded lettuce leaves. The drying rate curve represents the classic behavior of these type of drying rate curves, where its three stages are well defined—the induction period, constant drying rate, and falling rate period. Unlike what happened in the curves of drying and drying rates of the present investigation, in other aromatic plants, such as mint, parsley, and basil leaves dried directly in the sun, there was no period of constant speed in drying. The whole drying process occurred in the falling rate period, starting from the initial moisture content for mint, parsley and basil leaves (86, 84, 87 ± 0.5%, wet base) to the final moisture content (10, 6, 4 ± 0.5%, wet base), respectively [46]. Other under open sun drying experiments on thin layers of crain-crain, fever, and bitter leaves grown in Nigeria, also the drying process took place in the falling rate period, and no constant rate period was observed from the drying curves [49]. This also occurred for the forced convection drying of parsley leaves, where there was no constant rate drying period, and it was observed that all drying operations occurred in the falling rate period [48]. In another study conducted with thymus and mint in convective solar dryer indirect, it was observed that drying curves exhibited the constant and falling rate periods [50].

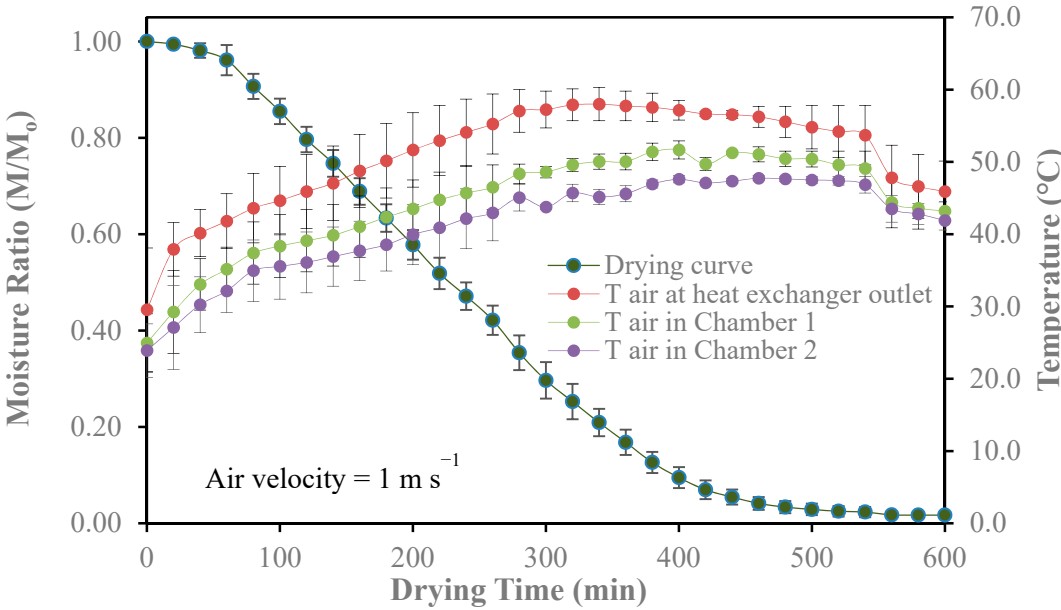

**Figure 2.** Relationship between the moisture ratio and the drying time and the air temperature versus drying time for the two drying chambers and heat exchanger outlet (mean values of the three runs performed).

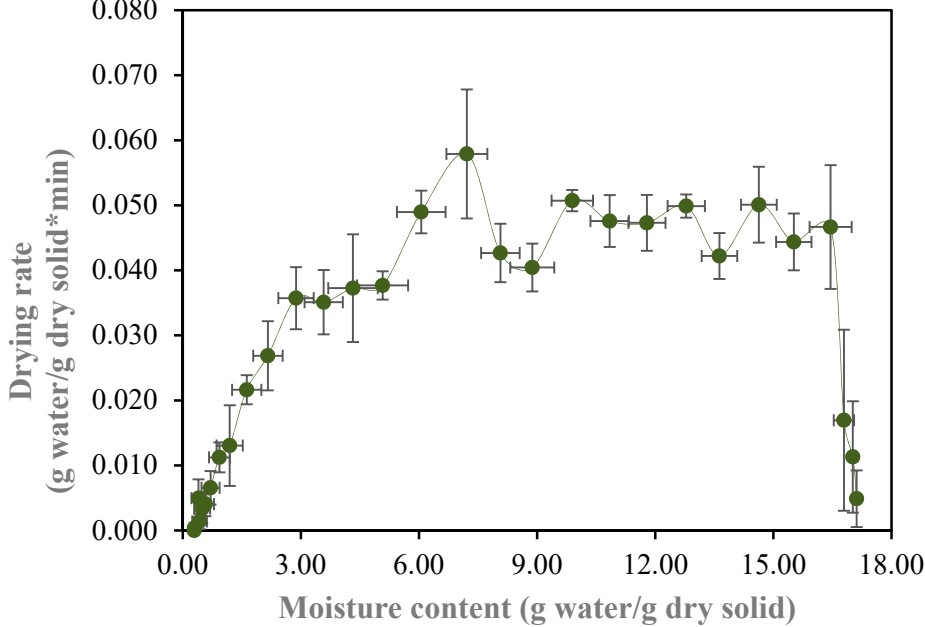

**Figure 3.** Drying rate versus moisture content from mean values of the two drying chambers (n = 3).

In Figure 3 the induction period begins with a value of the initial moisture content ($M_o$) of 17.16 g water/g dry solid, until reaching a moisture value of 16.45 g water/g dry solid at 60 min into the process where it arrives at an average drying rate of 0.0467 g water/g dry solid * min. At this stage, the air temperature reached mean values of 35.2 and 32.2 °C, chamber 1 and 2, respectively (Figure 2). From there the period of constant drying rate began to reach 280 min, remaining practically constant with average values of 0.0486 g water/g dry solid * min, and a mean value at the end of this stage for moisture content of 6.06 g water/g dry solid; this point in the curve is called critical moisture ($M_{crít.}$), this trajectory passed during 220 min; that is, 3 h and 40 min. It is at this point that the falling rate period begins, and the drying rate will progressively fall until equilibrium, reaching average values of

0.00012 g water/g dry solid * min and a final moisture, called equilibrium moisture ($M_{eq}$), with average values of 0.11 g water/g dry solid (Figure 3).

While the constant rate-drying period is carried out from the shredded lettuce leaves whose surface is saturated with water, the drying rate is controlled largely by the temperature of air, area of the exposed product, surface area product, and flow conditions. As expected, in all runs, the drying rate was always slightly higher in chamber 1. During the falling rate period, the drying rate decreases continuously with decreasing moisture content and increasing drying time. In this period, the material surface is less saturated with water, and the drying rate is controlled by diffusion of moisture from the interior of the material to the surface. This has been observed in other previous studies that have dried green leaves from plants [46,49].

### 3.2. Modeling of Drying Curves Statistical Parameters

Table 2 shows the results of the statistical parameters, $\chi^2$, SSE, RMSE and $R^2$, which were used to determine the goodness of fit of the drying curves of the fifteen models studied [40,42]. The criteria used for selecting the best fitting model was based primarily on the coefficient of determination ($R^2$) with values closer to 1.0, and values closer to zero for $\chi^2$, SSE, and RMSE. Thus, the best fit model, according to the thermal history displayed in the process drying curve for the group of the three models derived from Newton's law of cooling, Newton, Page, and Modified Page, was the Page model with values to $R^2$, $\chi^2$, SSE, and RMSE of 0.9989, $1.49 \times 10^{-4}$, $1.40 \times 10^{-4}$, and 0.01182, respectively. The values of the kinetic constant (k), and the empirical parameter (n) were: $1.53 \times 10^{-5}$ min$^{-1}$, and 1.98128, respectively. Amongst the tested models for the group of the nine models derived from Fick's second law of diffusion, Henderson and Pabis, Modified Henderson and Pabis, Logarithm, Approach of diffusion, Midilli and Kucuk, Two-Term, Two-Term Exponential, Aghbashlo, and Verma models, the one with the best fit was that Midilli and Kucuk model with values to $R^2$, χ2, SSE, and RMSE of 0.9988, $1.70 \times 10^{-4}$, $1.48 \times 10^{-4}$, and 0.01217, respectively; with empirical constants a, b, and n of, 1.00374, $-1.92 \times 10^{-5}$, 1.91162 and the kinetic constant (k) of $2.22 \times 10^{-5}$ min$^{-1}$, respectively.

Of the three tested models that belong to the group of so-called empirical models, the one that resulted in the best fit was the Weibull distribution model with values to $R^2$, $\chi^2$, SSE, and RMSE of 0.9985, $2.15 \times 10^{-4}$, $1.87 \times 10^{-4}$, and 0.01368, respectively; being the empirical constants a, b, and n of 1.01587, 0.00075, 1.88817 and the kinetic constant (k) of $2.60 \times 10^{-5}$ min$^{-1}$, respectively (Table 2).

Therefore, the three models that best represented the drying curve in this indirect solar drying installation with thermal energy storage in the form of sensible heat were: Page, Midilli and Kucuk, and Weibull. This implies a good correlation between the predicted humidity ratios obtained by these three models and the experimental humidity ratios to predict the drying kinetics of striped lettuce leaves. While these models can be used to predict changes in moisture content over time, they do not have the combined effect of the other parameters of drying, such as air velocity and drying temperature, since the present study is limited only for air velocity of 1 m/s and variable-drying temperatures, which did not exceed 51.7 °C.

Rojano-Aguilar et al. [20] demonstrated in a solar drying device tested with lettuce leaves, that by adjusting 16 thin-layer models of drying, the best fitting to be able to well-describe the drying kinetics was the "Two-Term" model. The difference of this result with those achieved in the present investigation is mainly due to the fact that the drying process was carried out with the lettuce leaves instead of using shredded lettuces leaves.

**Table 2.** Statistical results from modeling the moisture content and drying time for shredded lettuce leaves.

| N° | Models | Coefficients | | | | | | $R^2$ | $\chi 2$ | SSE | RMSE |
|---|---|---|---|---|---|---|---|---|---|---|---|
| 1 | Newton | $k = 3.85 \times 10^{-3}$ | | | | | | 0.8995 | $1.32 \times 10^{-2}$ | $1.28 \times 10^{-2}$ | 0.11302 |
| 2 | Page | $k = 1.53 \times 10^{-5}$ | $n = 1.98128$ | | | | | **0.9989** | **$1.49 \times 10^{-4}$** | **$1.40 \times 10^{-4}$** | **0.01182** |
| 3 | Modified Page | $k = 9.61 \times 10^{-3}$ | $n = 0.40023$ | | | | | 0.8995 | $1.37 \times 10^{-2}$ | $1.28 \times 10^{-2}$ | 0.11203 |
| 4 | Henderson and Pabis | $a = 1.20453$ | $k = 4.58 \times 10^{-3}$ | | | | | 0.9385 | $8.36 \times 10^{-3}$ | $7.82 \times 10^{-3}$ | 0.08842 |
| 5 | Modified Henderson & Pabis | $a = 0.26382$ | $b = 0.26382$ | $c = 0.67678$ | $k = 4.56 \times 10^{-3}$ | $g = 4.56 \times 10^{-3}$ | $h = 4.58 \times 10^{-3}$ | 0.9385 | $9.70 \times 10^{-3}$ | $7.82 \times 10^{-3}$ | 0.08842 |
| 6 | Logarithm | $a = 1.60883$ | $c = -0.48434$ | $k = 2.27 \times 10^{-3}$ | | | | 0.9781 | $3.08 \times 10^{-3}$ | $2.79 \times 10^{-3}$ | 0.05277 |
| 7 | Approach of diffusion | $a = -119.017$ | $b = 0.98918$ | $k = 9.31 \times 10^{-3}$ | | | | 0.9908 | $1.29 \times 10^{-3}$ | $1.17 \times 10^{-3}$ | 0.03414 |
| 8 | Midilli and Kucuk | $a = 1.00374$ | $b = -1.92 \times 10^{-5}$ | $n = 1.91162$ | $k = 2.22 \times 10^{-5}$ | | | **0.9988** | **$1.70 \times 10^{-4}$** | **$1.48 \times 10^{-4}$** | **0.01217** |
| 9 | Two Term | $a = 12.7062$ | $b = -11.7499$ | $k = 8.93 \times 10^{-3}$ | $k_1 = 9.99 \times 10^{-3}$ | | | 0.9915 | $1.24 \times 10^{-3}$ | $1.08 \times 10^{-3}$ | 0.03284 |
| 10 | Two Term Exponential | $a = 2.23266$ | $k = 6.67 \times 10^{-3}$ | | | | | 0.9868 | $1.85 \times 10^{-3}$ | $1.67 \times 10^{-3}$ | 0.04093 |
| 11 | Aghbashlo | $k_1 = 1.85 \times 10^{-3}$ | $k_2 = -1.66 \times 10^{-3}$ | | | | | 0.9928 | $9.80 \times 10^{-4}$ | $9.17 \times 10^{-4}$ | 0.03028 |
| 12 | Verma | $a = 14.6651$ | $k = 8.05 \times 10^{-4}$ | $g = 6.71 \times 10^{-4}$ | | | | 0.9650 | $4.93 \times 10^{-3}$ | $4.45 \times 10^{-3}$ | 0.06671 |
| 13 | Wang and Sing | $a = -2.64 \times 10^{-3}$ | $b = 1.42 \times 10^{-6}$ | | | | | 0.9678 | $4.37 \times 10^{-3}$ | $4.09 \times 10^{-3}$ | 0.06396 |
| 14 | Thompson | $a = -180.062$ | $b = 0.41438$ | | | | | 0.9378 | $2.76 \times 10^{-2}$ | $2.58 \times 10^{-2}$ | 0.16060 |
| 15 | Weibull Distribution | $a = 1.01587$ | $b = 0.00746$ | $k = 2.60 \times 10^{-5}$ | $n = 1.88817$ | | | **0.9985** | **$2.15 \times 10^{-4}$** | **$1.87 \times 10^{-4}$** | **0.01368** |

### 3.3. Model Validation

The consistency, adequacy, and reliability of these three models were substantiated by plotting the calculated moisture ratios obtained from the models against the experimental moisture ratio data, as shown in Figures 4–6, respectively. Hence these models are adequate, reliable, and suitable for predicting the drying kinetics of shredded lettuce leaves using solar drying in an indirect-mode.

It was clearly visible that the established models provided a good correlation between the experimental and predicted *MR*s since the predicted data forms a straight line at 45°. Thus, it can be said that the models are valid for describing the thin-layer drying behavior of shredded lettuce leaves under the given experimental conditions.

The empirical equations obtained from thin layer drying models of shredded lettuce leaves dried in an indirect mode solar dryer can be expressed according the Page, Midilli and Kucuk, and Weibull models (Equations (17)–(19)):

$$MR = exp\left(-0.0000153\ t^{1.98128}\right) \tag{17}$$

$$MR = 1.00374\ exp\left(-0.0000222\ t^{1.91162}\right) + \left(-0.0000192\ t\right) \tag{18}$$

$$MR = 1.01587 - 0.00746\ exp\left(-0.000026\ t^{1.88817}\right) \tag{19}$$

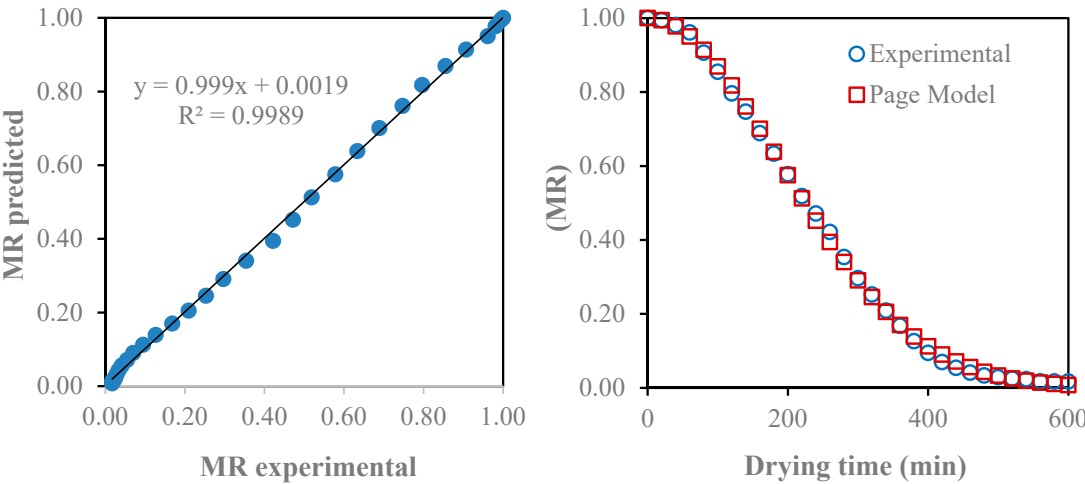

**Figure 4.** Validation of experimental and predicted moisture ratio values from the Page model.

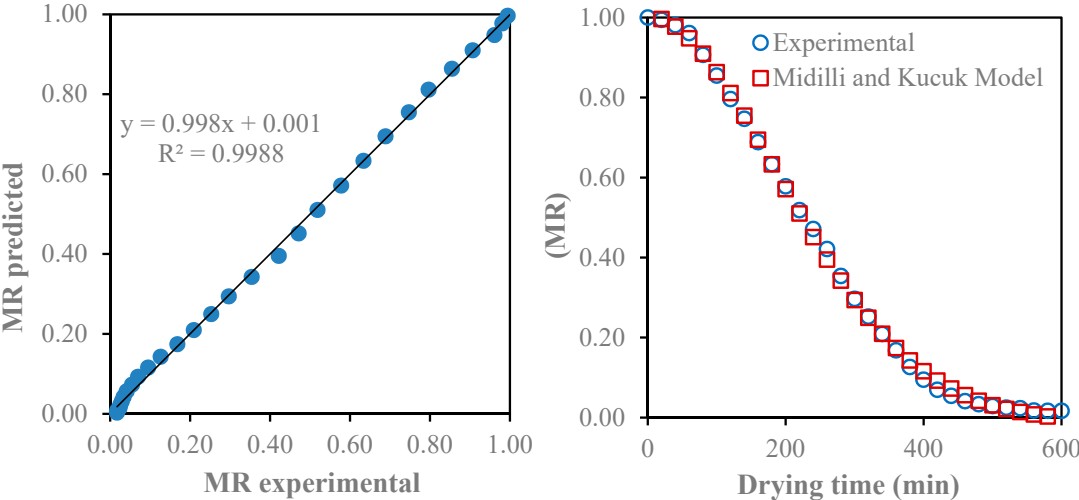

**Figure 5.** Validation of experimental and predicted moisture ratio values from the Midilli and Kucuk model.

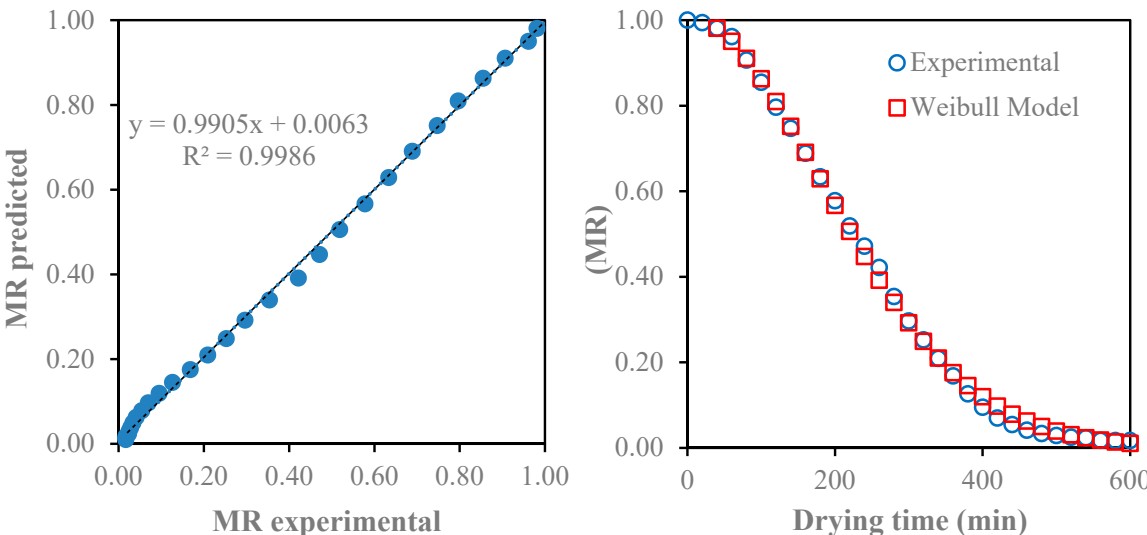

**Figure 6.** Validation of experimental and predicted moisture ratio values from the Weibull Distribution model.

### 3.4. Chromatic Coordinates

The values of brightness $L^*$ and the chromaticity coordinates $a^*$ and $b^*$ to the fresh and dried sample of lettuce are shown in Table 3. After the drying process was completed it was observed that the values of $L^*$ between fresh and dried samples had no statistically significant differences ($p < 0.05$), indicating that the drying process did not affect this parameter. However, a different behavior was taken to the values of $a^*$ and $b^*$ where statistically significant differences ($p < 0.05$) were recorded to compare the values of the population of the fresh sample with the dry sample. Chromatic coordinate $a^*$ had a move from the green zone to the beginning of the red zone, which is indicative of the change of sign from negative to positive, while in the case of the chromatic coordinate $b^*$, although the values decreased, without significant change, they remained in the first quadrant of the sphere of color, encouraging a decrease of yellow color.

A color difference ($\Delta E$) was observed in the drying process, between fresh and dried samples, which reached a mean value of $\Delta E = 8.26$ (Table 3). This color difference represented a perceptible change of color, higher than that considered for a product that initiates the instability color ($\Delta E \geq 5.0$) indicated by Obon et al. [51] were also higher than $\Delta E \leq 3.0$, the upper limit for the values that cannot be easily detected by the human eye with normal vision, according to the findings of Hong et al. [52].

In addition, the ratio $R$ (redness ratio in yellow) (Table 3), which is an index of high applicability to study the possibilities of browning in foodstuffs [53], increased slightly from the initial mean values for the fresh sample $R = -0.20$ to the dry sample of $R = 0.05$, due to an increase in the values of the chromatic coordinate to $a^*$, from the green zone ($a^* = -$) to the red zone ($a^* = +$) of the sphere of color, and a decrease in the average of the values of the chromatic coordinate $b^*$. The fresh samples had the value of hue-angle ($h° = 101.43 \pm 0.15$) belonging to the area of the second yellow-green quadrant of the color sphere, and passed to the area of the first quadrant of the color sphere with hue-angle ($h° = 87.16 \pm 0.41$) for dried samples, which is an expected color in dried shredded lettuce leaves at low temperatures (<55 °C). On the other hand, the Chroma ($C$), an indicator of the intensity and saturation of the color; their mean values of the fresh samples, $C = 22.72 \pm 0.11$, were higher than those of the dried sample, $C = 16.07 \pm 0.32$ (Table 3), for a $\Delta C = 6.65$, indicating a "dull color". This is an expected color condition for maximum drying temperatures reached ($\leq 52$ °C), and drying times of ~10 h.

The average browning index ($BI$) value of fresh lettuce before drying was $BI = 120.5 \pm 0.70$, and at the end of the process the value decreased to $BI = 78.99 \pm 0.50$, is a statistically significant difference ($p < 0.05$), which is equivalent to 65.55% of the initial value. This affirms that the drying process, although it is carried out indirectly without receiving the solar incidence, this parameter in the dry

sample was affected. It is proposed that the color changes from bright green to olive-brown occur in green leafy vegetables, such as mustard leaves (*Brassica campestris*), spinach (*Spinacea oleracea* L.), and fenugreek (*Trigonella foenum* L.) leaves during thermal processing, and are accompanied by the conversion of chlorophyll into pheophytin [54].

**Table 3.** Chromatic Coordinates of Lettuce in Fresh ($n = 10$) and after Drying process ($n = 20$). Color Difference ($\Delta E$), Ratio of redness over yellowness ($R$), and Browning Index ($BI$) ($n = 3$), fresh and after the drying process.

| Chromatic Coordinates and Others Parameters | Samples | |
|---|---|---|
| | Fresh | Dried |
| $L^*$ | $27.97 \pm 0.05$ [a] | $28.92 \pm 0.71$ [a] |
| $a^*$ | $-4.5 \pm 0.07$ [a] | $0.8 \pm 0.13$ [b] |
| $b^*$ | $22.27 \pm 0.10$ [a] | $16.5 \pm 0.32$ [b] |
| $\Delta E$ | - | $8.26 \pm 0.18$ |
| $R = \frac{a^*}{b^*}$ | $-0.20 \pm 0.00$ [a] | $0.05 \pm 0.01$ [b] |
| $h^\circ$ | $101.43 \pm 0.15$ [a] | $87.16 \pm 0.41$ [b] |
| $C$ | $22.72 \pm 0.11$ [a] | $16.07 \pm 0.32$ [b] |
| $BI$ | $120.50 \pm 0.70$ [a] | $78.99 \pm 0.50$ [b] |

Mean values in the same row with different letters differ significantly ($p \leq 0.05$).

### 3.5. Effective Moisture Diffusivity ($D_{eff}$)

The $D_{eff}$ values of shredded lettuce leaves obtained from Equation (15) is shown in Figure 7 in which the ln ($MR$) is related to the drying time in order to obtain the slope $K$ and from there perform the $D_{eff}$ determination.

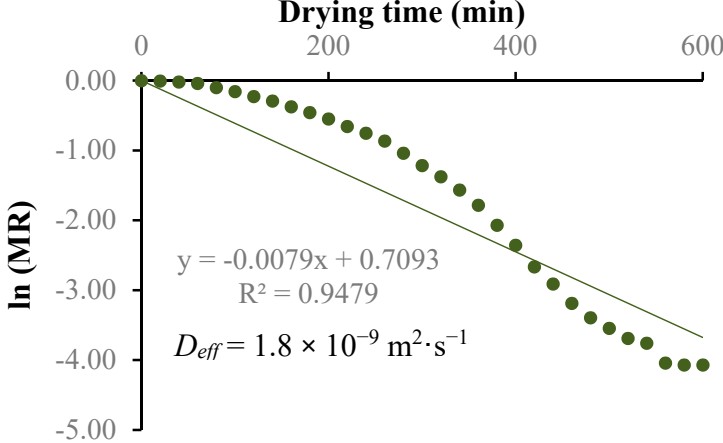

**Figure 7.** Linear relationship between ln(MR) and drying time for temperature value < 52 °C.

$D_{eff}$ value for shredded lettuce leaves of $1.8 \times 10^{-9}$ m$^2$·s$^{-1}$ was reached for temperature values $\leq 52$ °C. $D_{eff}$ values for lettuce leaves has not been found in the literature, but the following values have been reported for other vegetable leaves in thin-layer drying for vegetable waste (as a mixture of lettuce and cauliflower leaves) from the wholesale market for a temperature range of 50–150 °C, $D_{eff}$ values varied from $6.03 \times 10^{-9}$ to $3.15 \times 10^{-8}$ m$^2$ s$^{-1}$ [19], demonstrating that $D_{eff}$ values are higher with increasing temperature. In the present solar dryer indirect, although the shredded lettuce leaves were subjected to a constant airspeed (1 m·s$^{-1}$), its temperature throughout the drying process was rising, starting at 25 °C until reaching a maximum of 51.7 and ending at 41 °C, so that the value of the $D_{eff}$ is the result of a mean value obtained by fitting the function shown in Figure 7 to a straight line.

These $D_{eff}$ values correspond to similar values obtained for other green leaves vegetables, herbs, and aromatic plants, for example, the thin-layer drying behavior of mint leaves for a temperature range of 35–60 °C, the $D_{eff}$ varied from $3.07 \times 10^{-9}$ to $1.94 \times 10^{-8}$ m$^2$ s$^{-1}$ and increased with the air temperature [54,55]. In another study on the thin layer drying of mint leaves [55,56], $D_{eff}$ values between $1.23 \times 10^{-10}$ and $2.66 \times 10^{-10}$ m$^2$ s$^{-1}$ were reported for a temperature range of 45 to 65 °C. Doymaz et al. [56,57] determined that the $D_{eff}$ values for 50, 60 and 70 °C were $6.69 \times 10^{-10}$; $9.205 \times 10^{-10}$, $1.434 \times 10^{-9}$ m$^2$ s$^{-1}$ for dill leaves, and $9.0 \times 10^{-10}$, $1.36 \times 10^{-9}$, $2.35 \times 10^{-9}$ m$^2$ s$^{-1}$ for parsley leaves, respectively.

## 4. Conclusions

The aim of the present study, related to the drying of chopped lettuce leaves (*Lactuca sativa* L. var. longifolia), in a pilot solar dryer of indirect operation, using thermal energy storage in water and equipped with solar collectors to convert solar irradiance into thermal energy, and photovoltaic cells to convert solar irradiance into electricity, and with partial recovery of the outlet air, was fulfilled, achieving a relatively short time (=10 h) considering that in other types of convective solar dryers without energy storage thermal, the times would be longer. Of the fifteen thin-layer drying models studied, only three of them: Page, Midilli and Kucuk, and Weibull Distribution, were the ones that had the best fit according to the established statistical parameters. The product obtained dry, although it had a more opaque and less bright color than the fresh product, its quality for this type of process was appropriate. In future research, this drying facility will continue to be perfected in which higher airspeed variations can be made, allowing shorter drying times. In addition, due to its thermal energy storage condition, the dryer can be operated at night or on cloudy days.

**Supplementary Materials:** The following are available online at http://www.mdpi.com/2227-9717/8/2/168/s1, Some aspects of interest of the drying vehicles in the solar installation. Water and air/Figure S1: Water temperature in different places of the circuit performed by the solar drying installation, Figure S2: Inlet and outlet air temperature in the solar drying installation.

**Author Contributions:** Conceptualization, P.C.M. and A.Á.L.; Data curation, P.C.M. and A.Á.L.; Formal analysis, A.Á.L. and W.B.M.; Funding acquisition, P.C.M.; Investigation, P.C.M., A.Á.L., and W.B.M.; Methodology, P.C.M., and A.Á.L.; Resources, P.C.M.; Software, P.C.M., A.Á.L., and W.B.M.; Supervision, P.C.M.; Writing—original draft, P.C.M., A.Á.L., and W.B.M.; Writing—review & editing, P.C.M. All authors have read and agreed to the published version of the manuscript.

**Funding:** This research was funded by two projects with public funds: Competitive Regional Innovation Fund of Antofagasta - Chile, (Grant: FIC-R 4602) and by the University of Antofagasta through of the Seedbed Project No. 4467 entitled: "Solar Energy Available to the New Technological Challenges in the Drying of Biological Products".(Grant: Seedbed Project – 4467).

**Acknowledgments:** To the Research Management Directorate of the University of Antofagasta by grant undergraduate scholarship fund to Aldo Alvarez López to carry out this research. Finally, we want to give special recognition to Carlos Dario Alfonso for his help in translation.

**Conflicts of Interest:** The authors declare no conflict of interest.

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
