# Peer review of "Effect of Drying on Lettuce Leaves Using Indirect Solar Dryer Assisted with Photovoltaic Cells and Thermal Energy Storage"

_processes, doi:10.3390/pr8020168_

Round 1

Reviewer 1 Report

Abstract. Some conclusions are missing. For example which was the general result of the study? Any possible application? Line 13. “Main results:” I do not like this way of presentation. It would be better to make sentences. Lines 41-51. Delete. There is no correlation with the present study. Line 85. No meaning to know where the market is. Delete. Line 91. Please do not present results in the materials and method section. Figures should be numbered according to first appearance in the text. Please revise. Lines 412-437. I cannot understand the meaning and the impact of these section. In the best scenario it may be presented as supplementary material. Tables. Use . in all tables. Please correct , In general, the work is interesting, however, the presentation has many issues. Several parts in the results and discussion have no meaning and may be easily deleted. The text should be reduced and the authors should present the most important results. Some information should be presented as supplementary material. There is limited discussion with the results of other studies. For example a similar study with solar drying of lettuce [22] is only discussed one time. The authors should present clearly in the text the aim, originality and importance of their study. In the conclusions, the authors should not present again the results but the importance of these results and possible applications in the industry or potential future works. Final comments. 1. Reduce introduction, 2. Reduce the results and discussion (several parts should be deleted or as supplementary material, 3. Present originality, novelty aim of the work, 4. Rewrite conclusions.

Reviewer 2 Report

The topic of the manuscript is of scientific relevance since it describes the drying using indirect solar dryer assisted with photovoltaic cells.

This study provides new knowledge about  drying of shredded lettuce (Lactuca sativa L. var. Longifolia) leaves, in a pilot solar dryer.

The description of the methods shows a sound scientific approach.

There is no discussion in this paper, authors just gave some examples from other studies. Language grammar should be revised by an expert language editor.

Some revision are highlighted in the manuscript.

Some questions are indicted in the manuscript which should be cleared.

Round 2

Reviewer 1 Report

The authors replied in all my comments and I think that the manuscript has been improved and may be accepted.